# PLPP: Prompt Learning with Perplexity for Vision-Language Models

## Abstract

Pre-trained vision-language (VL) models such as CLIP have demonstrated their excellent performance across numerous downstream tasks. A recent method, called Context Optimization (CoOp), further improves the performance of CLIP on downstream tasks by introducing prompt learning. CoOp optimizes a set of learnable vectors, aka prompt and freezes the whole CLIP model, instead of using manually crafted templates (e.g., a template "a photo of a {category}") to fine-tune the CLIP model. Nonetheless, we observed that the resulting prompts are always incomprehensible, which is counter-intuitive, and existing CoOp-based methods overlook this issue. As the first work aiming at learning comprehensible prompts, this paper proposes to use Perplexity to supervise the process of prompt learning in the CoOp framework. Perplexity is a metric to evaluate the quality of a language model (LM) in Natural Language Processing field, and we design a two-step operation to compute the perplexity for prompts. The first step is a calculation of cosine similarity to obtain the labels of vectors, and the second step is a training-free LM Head to output word probability distribution. Our proposed method, i.e., **P**rompt **L**earning with **P**er**P**lexity (PLPP), can be integrated in any CoOp-based method and the experiments show that the learned prompts are much more comprehensible compared with the original and an improved CoOp methods, without sacrificing model accuracy. Codes are available at https://github.com.

## 1 Introduction

In the past few years, the advent of CLIP Radford et al. (2021) and ALIGN Jia et al. (2021) has sparked heightened interest in the exploration of vision-language (VL) models endowed with the capability to engage in integrated reasoning, utilizing both visual and textual information. It is worth noting that such models exhibit a voracious appetite for data, necessitating training on an extensive corpus of image-text pairs. For instance, CLIP's training regimen involves a staggering 400 million image-text pairs. Following the pre-training phase, VL models can perform image classification by employing a carefully crafted prompt, such as "a photo of a {category}," as input for the text encoder. Simultaneously, the image encoder processes the visual input. The ultimate classification results are then derived by computing the embeddings of both the image and text representations across all categories.

While the development of high-quality contextual prompts Jin et al. (2022) has demonstrated its capacity to enhance the performance of CLIP and other similar VL models, it often relies upon a considerable expenditure of time and the specific domain knowledge of human experts. This resource-intensive process may also exhibit limited efficacy when confronted with novel or unforeseen scenarios. Furthermore, the combination of a vast parameter space and constraints on available training data, particularly in a few-shot setting, renders it unfeasible to perform comprehensive fine-tuning of the entire model for downstream tasks.

Engaging in such fine-tuning carries the added risk of erasing valuable knowledge acquired during the large-scale pretraining phase and introducing the potential for overfitting to the specific downstream task. To address these challenges, inspired by recent advances in Natural Language Processing (NLP) Gao et al. (2021b); Jiang et al. (2020); Lester et al. (2021); Li & Liang (2021); Shin et al. (2020); Zhong et al. (2021), CoOp Zhou et al. (2022b) introduces a prompt learning methodology as an alternative to manually crafting prompts for specific tasks. Diverging from prior fine-tuning

paradigms, CoOp keeps both the image and text encoders of CLIP fixed, exclusively fine-tuning the prompt, which consist of a set of randomly initialized vectors. Following in the footsteps of CoOp Zhou et al. (2022b), several approaches have been proposed to enhance the training paradigm for prompt or to introduce a novel learnable prompt, as exemplified by Hantao Yao (2023); Zhou et al. (2022a); khattak et al. (2023); Lu et al. (2022); Zhu et al. (2023); Chen et al. (2023); Xing et al. (2023). However, these approaches have predominantly concentrated on improving performance across a spectrum of downstream tasks, often ignored that the resulting prompt is utterly incomprehensible. Aiming at learning comprehensible prompts and maintaining high accuracy of

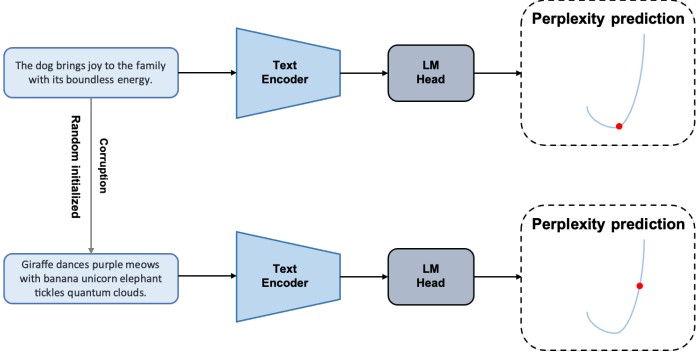

Figure 1: Overview of perplexity prediction to two different sentences. Red pot represents the value of perplexity. Low perplexity prediction is assigned to a hand-written sentence, while high perplexity prediction is assigned to a sentence composed of randomly initialized words.

CoOp on downstream tasks, we propose to use Perplexity to supervise the process of prompt learning in CoOp, called Prompt Learning with PerPlexity (PLPP). Our motivation is straightforward. In energy-based models (EBMs) Wang et al. (2023), the energy function assigns low/high energy score to labeled/unlabeled data. In Natural Language Processing Pillutla et al. (2021), the perplexity measures the quality of a language model. Therefore, perplexity can be regarded as the energy function in prompt learning by supposing the text encoder of CLIP is a well-trained language model.

In summary, our main contributions of this work are as follows:

- To the best of our knowledge, this paper is the first endeavor to explore the comprehensibility of prompt in the context of vision-language models without performance losing since readable prompts helps in understanding the predication results and applying the VL models in safety-critical scenarios.

- Through incorporating perplexity into the loss function for learnable prompt, we propose a novel CoOp-based method, called Prompt Learning with PerPlexity (PLPP). In PLPP, the perplexity of prompt is calculated by feeding the prompt label (measured by cosine similarity) and the corresponding word probability distribution (the output of a training-free LM Head) into the cross-entropy function.

- • PLPP can be integrated in any CoOp-based method, and we conduct extensive experiments across different task settings, such as few-shot classification, base-to-new generalization and domain generalization, to demonstrate its effectiveness. Compared with CoOp and CoCoOp, experimental results show that the learned prompts by PLPP contain more actual words and the obtained VL models have a competitive accuracy rate at the same time.

## 2 RELATED WORKS

**Pre-training for VL models.** Owing to voracious appetite of vision-languange (VL) models for data, the pre-training stage for vision-language (VL) models entails unsupervised learning on a substantial dataset prior to the model's official deployment for a specific task. The goal of this phase is to facilitate the model to align the features of the image with the corresponding text features.

CLIP Radford et al. (2021) and ALIGN Jia et al. (2021) utilize more than four million image-text pairs for pre-training. To ensure the proximity of analogous inputs within the same modality, TCL Yang et al. (2022) employs a combination of cross-modal and intra-modal self-supervision, yielding synergistic advantages in representation learning. In a concerted effort to bolster training efficiency, DeCLIP Li et al. (2022) not only exploits cross-modal multi-view and intra-modal supervision but also introduces a novel cross-modal Nearest-Neighbor Supervision mechanism, which taps into information emanating from analogous pairs in a more nuanced manner. OneR Jang et al. (2023) and MS-CLIP You et al. (2022) both adopt a unified transformer encoder architecture for image-text pairs. To explicitly capture the hierarchical essence of high-level and fine-grained semantics embedded in both images and textual content, HiCLIP Geng et al. (2023) enhances the visual and textual encoders of CLIP with hierarchy-aware attentions, enabling the model to learn semantic hierarchies in a layer-by-layer fashion.

**Enhancement of Modules in Pre-trained VL Models.** CALIP Guo et al. (2023) introduces an ingenious attention module devoid of parameters, thereby augmenting the zero-shot performance of CLIP. CALIP orchestrates the harmonious interplay between visual and textual representations, delving into cross-modal informative features through the medium of attention, all accomplished without incurring additional training expenses or heightened data requirements. Furthermore, CALIP harnesses the latent potential of cross-modal interactions in a few-shot scenario, a feat achieved by the incorporation of several trainable linear layers both preceding and succeeding the attention module. CLIP-Adapter Gao et al. (2021a), on the other hand, introduces an additional bottleneck layer into the model, which undertakes the acquisition of novel features and executes residual-style feature fusion with the originally pretrained features. This strategic addition enables the model to gracefully adapt to novel tasks, safeguarding against overfitting while retaining the advantages conferred by its pretrained foundation. Meanwhile, Tip-Adapter Zhang et al. (2022) inherits the advantageous feature of being training-free, as seen in CLIP-Adapter. It further elevates its downstream task performance by generating weights through the creation of a key-value cache model derived from the few-shot training set, thus enhancing its adaptability and efficacy.

**Leanable Prompt for Pre-trained VL Models.** Prompt learning represents a recent stride in the realm of Natural Language Processing (NLP). CoOp Zhou et al. (2022b) stands as the pioneering endeavor to employ such a method in the customization of expansive vision-language models within the domain of computer vision. This innovation yields substantial improvements in performance when juxtaposed with manually crafted prompts in downstream tasks, particularly in the realm of few-shot classification. Nevertheless, CoOp exhibits constraints in its aptitude to generalize across broader, unseen classes within the same dataset. In response, CoCoOp Zhou et al. (2022a) extends this paradigm by cultivating a nimble neural network, tasked with generating an input-conditional token for each image, thus amplifying its generalization capabilities. Models grounded in CLIP's architecture still rely upon manual prompts for image classification, thereby encumbering their ability to fully harness the vast reservoir of knowledge harbored within the CLIP text encoder. To surmount this limitation, Prompt-Adapter Sun et al. (2023) melds pre-trained prompt fine-tuning with an efficient adaptation network, culminating in superior few-shot classification performance. The utilization of prompting within a solitary branch of CLIP is suboptimal, as it constrains the model's adaptability to adjust both representation spaces for downstream tasks. Enter MaPLE khattak et al. (2023), which pioneers prompt learning in both the visual and language branches, effecting a tangible enhancement in the alignment of representations. Additionally, MaPLE introduces a profound prompting strategy that extends the purview of prompt learning not solely to the input but across multiple transformer blocks. DPT Xing et al. (2023) propounds a groundbreaking paradigm, one that concurrently imbibes the erudition of text and visual prompts. Furthermore, it advances the Class-Aware Visual Prompt Tuning (CAVPT) scheme, dynamically engendering visual prompts based on both task-related and instance-specific cues. When CoOp-based methodologies are employed in the training of downstream tasks, leanable prompts tend to accrue task-specific textual knowledge but tend to overlook the pivotal reservoir of general textual knowledge that underpins robust generalization. KgCoOp Hantao Yao (2023) intervenes to mitigate the divergence between the textual embeddings generated by learned prompts and their hand-crafted counterparts, averting the loss of essential knowledge.Moreover, ProGrad Zhu et al. (2023) introduces a selective update mechanism for prompts, exclusively attending to those prompts whose gradients align with the gradients of the Kullback-Leibler (KL) loss, calculated by reconciling learnable prompts and hand-crafted prompts. This alignment criterion necessitates that the angle between the two kind of gradients falls below $90°$.

## 3 METHODOLOGY

In this section, we provide a all-encompassing overview of CoOp Zhou et al. (2022b). Additionally, we introduce our innovative method, **P**rompt **L**earning with **P**er**P**lexity (PLPP), which aims to enhance the comprehensibility of prompt with the assistance of perplexity in training while maintaining comparable performance in downstream tasks.

### 3.1 A OVERVIEW OF COOP

CoOp Zhou et al. (2022b), originally conceived to bolster the performance of CLIP Radford et al. (2021) on few-shot and domain generalization tasks, heralded a pivotal shift by introducing prompt learning to vision-language (VL) models. Instead of using hand-crafted prompt templates, CoOp initialize a set of learnable vectors, each vector dimension is 512, which is consistent with the dimension of word embeddings. The number of vectors is typically set to values such as 4, 8, or 16. Concretely, the learnable vector set is denoted as $V = \{v_1, v_2, \ldots, v_M\}$, with $M$ being the count of vectors. Each prompt, denoted as $p_i = \{v_1, v_2, \ldots, v_M, c_i\}$, amalgamates these learnable vectors with the class token embedding $c_i$, where $c_i$ represents the tokenized class name corresponding to the $i$-th class. Subsequently, all prompts are feed into CLIP's text encoder, denoted as $g(.)$. Assuming $f$ represents the visual embedding of $x$, the ultimate prediction probability for predicting the image $x$ as $i$-th class is calculated as follows:

$$p(y = i|x) = \frac{exp(sim(g(t_i), f)/\tau)}{\sum_{j=1}^{K} exp(sim(g(t_j), f)/\tau)},\tag{1}$$

where $sim(.,.)$ signifies a metric function such as cosine similarity, and $\tau$ corresponds to the Softmax temperature. Finally, given an image and its label, the prediction probability and the labeled target are utilized to compute the cross-entropy loss, optimizing the learnable vectors $V$, while the parameters of the text and image encoder are frozen.

### 3.2 PERPLEXITY

Perplexity Pillutla et al. (2021) serves as a prominent metric used to aseess the quality of a language model, quantifying its capacity to predict a given sequence. At its core, a Language Model (LM) strives to output a probability distribution over a predefined vocabulary of words. Consequently, when subjected to test sets comprising hand-crafted sentences, a higher probability assigned to the corresponding output signifies a superior LM, while conversely, a lower probability suggests otherwise. For a given sentence in the test set, denoted as $W = \{w_1, w_2, \ldots, w_N\}$, where $N$ signifies the total sentence length, perplexity of the sentence is calculated as follows:

$$\begin{aligned} Perplexity(W) &= p(W)^{-\frac{1}{N}} \\ &= e^{-\frac{1}{N} \sum_{i=1}^{N} \log p(w_i|w_{<i})} \\ &= e^{H(q,p)} \end{aligned}\tag{2}$$

A detailed calculation of perplexity is as Equation 2. Here, $p(w_i|w_{<i})$ represents the probability of a word appearing at the $i$-th position in a sentence, with the specific word index being denoted earlier, i.e. model prediction distribution. The index $q$ corresponds to the indices of these words in $vocab\_size$ which maps words to integers, i.e. ground truth and $H$ signifies the cross-entropy function. It's worth noting that in the realm of Natural Language Processing (NLP), perplexity is primarily employed to evaluate the efficacy of a language model when exposed to human-authored sentences. In the context of our approach, we assume that our language model is well-trained, and the input text, albeit disordered, is amenable to learning. By minimizing perplexity, our objective is to guide the learnable text towards a more human-comprehensible orientation while preserving performance parity.

### 3.3 PROMPT LEARNING WITH PERPLEXITY

In this subsection, we delve into the intricacies of PLPP, a novel approach that bridges the gap between the evaluation metric perplexity Pillutla et al. (2021) and the realm of prompt learning in

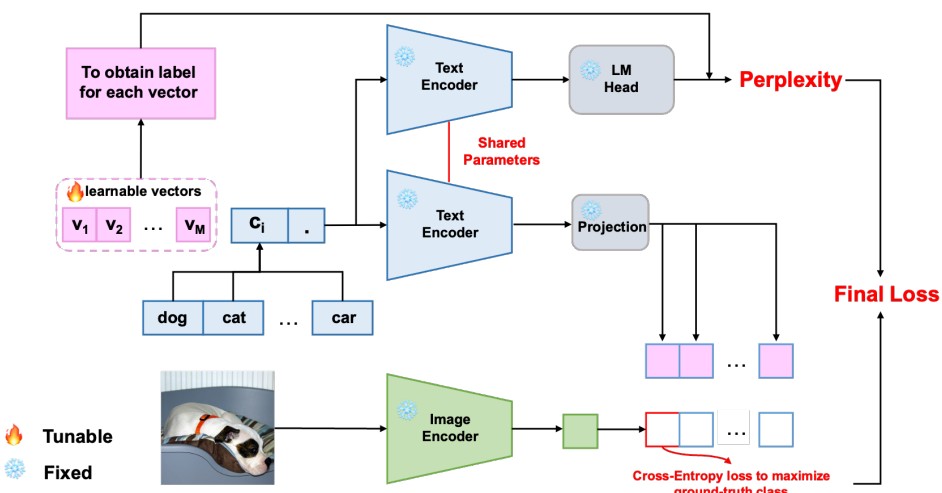

Figure 2: Overview of PLPP. In order to integrate perplexity in training process, we obtain the labels of vectors by calculating consine similarity and add a training-free LM Head to output word probability distribution, and then calculate the perplexity of prompt. Finally, optimizing the prompt together with the alignment loss.

vision-language models. The flow chart of our PLPP is as shown in Fig 2. Specifically, we first randomly initialize the leanable vectors $V = \{v_1, v_2, ..., v_M\}$, and the input for text encoder of CLIP is $p_i = \{v_1, v_2, ..., v_M, c_i\}$ where $i$ is related to the number of categories in the dataset. Given an image $x \in R^{H \times W \times 3}$ and its ground truth label $y \in Y$, we feed x into the image encoder of CLIP to get the visual embedding $f \in R^d$. $g(p_i) \in R^d$ is regarded as the text embedding of $p_i$ where $i$ represents the index of category. Then we calculate prediction probabilities for all the categories according to Equation 1. Subsequently, the cross-entropy loss is based on the prediction probabilities and ground-truth label $y$ for image $x$.

As for perplexity, we introduce an additional LM Head module positioned after the text encoder of CLIP to output word probability distribution, requiring no training. The LM Head comprises a straightforward linear layer devoid of bias, with its weight initialized using the transpose of $embedding.weight$, referencing the weight parameter of the embedding layer. Given that perplexity can be expressed in terms of cross-entropy loss and necessitates the labels of prompts, we utilize the dot product to calculate the cosine similarity between $V$ and the weight parameter of the embedding layer. This operation returns the index corresponding to the maximum similarity as the label for the prompts. After each batch, when prompts are updated, the label assignments for the prompts need to be recalculated. The pseudo code of our PLPP is shown in Appendix.

In summary, we denote $L_{CE}$ and $L_{PPL}$ as the loss function of as the loss of aligning the image-text features and perplexity loss of learnable promts. $\lambda$ controls the weights of perplexity loss. We have the overall loss function of PLPP as in Equation. 3

$$L_{PLPP} = L_{CE} + \lambda L_{PPL} \tag{3}$$

# 4 EXPERIMENTS

We evaluate the performance of PLPP across three distinct experimental configurations for the image recognition task: (1) few-shot classification (refer to Section 4.2), (2) base-to-new generalization (refer to Section 4.3), (3) domain generalization (refer to Section 4.4), and (4) visualizing learned prompts (refer to Section 4.5).

### 4.1 DATASETS AND IMPLEMENTATION DETAILS

**Datasets.** In few-shot learning and base-to-new generalization, we adhere to the methodology established by CoOp Zhou et al. (2022b) and CoCoOp Zhou et al. (2022a), utilizing a total of 11 image classification datasets to comprehensively evaluate the efficacy of our approach. These encompass two general object datasets, namely ImageNet Deng et al. (2009) and Caltech101 Fei-Fei et al. (2004), along with five fine-grained image recognition datasets, specifically OxfordPets Parkhi et al. (2012), StanfordCars Krause et al. (2013), Flowers102 Nilsback & Zisserman (2008), Food101 Bossard et al. (2014), and FGVCAircraft Maji et al. (2013). Additionally, we incorporate a satellite-image dataset, EuroSAT Helber et al. (2019), an action classification dataset, UCF101 Soomro et al. (2012), a texture classification dataset, DTD Cimpoi et al. (2014), and a scene recognition dataset, SUN397 Xiao et al. (2010), into our evaluation repertoire. In the context of domain generalization, we designate ImageNet as the source dataset, while we assess the performance across a spectrum of target datasets, including ImageNetV2 Recht et al. (2019), ImageNet-Sketch Wang et al. (2019), ImageNet-A Hendrycks et al. (2021b), and ImageNet-R Hendrycks et al. (2021a).

**Implementation Details.** In few-shot learning, we follow the methodology outlined in CoOp by training our PLPP model with 1, 2, 4, 8, and 16 shots by two version: for both unified vectors and class-specific vectors, positioning the class token in the end. Subsequently, conducting evaluations on the test dataset. In both domain generalization and base-to-new generalization, to further demonstrate the effectiveness of our method, we assess 16-shot performance compared with both CoOp Zhou et al. (2022b) and CoCoOp Zhou et al. (2022a). To ensure equitable comparisons, we compute the results for all methods and models by averaging over three random seeds. For few-shot classification, we adhere to the guidelines provided in CoOp, utilizing ResNet-50 He et al. (2016) as the backbone for the image encoder. For domain generalization and base-to-new generalization, we use vit-b/16 Dosovitskiy et al. (2020) as the image encoder. The number of learnable vectors, denoted as M, is consistently set to 16. We maintain congruity with the training epochs, training schedule, and data augmentation settings of CoOp and CoCoOp. For all settings, we set $\lambda$ to 1. Due to the experimental model is half precision, perplexity can give rise to issues of overflow, we thus substitute for $e^{H(q,p)}$ with $H(q,p)$ according to Equation 2.

**Baselines.** In our comparative analysis of few-shot classification, we juxtapose PLPP against four baseline approaches. The first baseline, Zero-shot CLIP Radford et al. (2021), relies on manually constructed prompts, with prompt design aligned with the specifications delineated in CoOp. The second baseline, Linear probe CLIP Radford et al. (2021), involves training a linear classifier after the CLIP image features. The third baseline, CoOp Zhou et al. (2022b), learning the unified context prompt through data-driven means rather than relying on manual design. Lastly, the fourth baseline, learning the class-specific context prompt. In our comparative analysis of base-to-new generalization and domain generalization, we use the nified context prompt of CoOp as the first baseline approach. CoCoOp extends CoOp by assimilating image-conditional prompt into its framework, a departure from a static prompt, thereby enhancing generalization capabilities. To further verify the effectivess of PLPP, we use CoCoOp as the second baseline approach.

### 4.2 FEW-SHOT CLASSIFICATION RESULTS

Figure 3 elucidates the comparative analysis across 11 diverse datasets. In an overarching perspective, our PLPP manifests distinct advantages over the baseline models across various few-shot scenarios, showcasing light improvement in average performance compared with CoOp Zhou et al. (2022b). Concretely, in StanfordCars, Flowers102, FGVCAircraft, SUN397, DTD, EuroSAT and UCF101 datasets, PLPP shows very close performance for all shots circumstances, including prompt is unified or class-specific. For imagenet, PLPP achieves 2.72% improvement for 1 shot, unified prompt, and for other situations, PLPP continuously lead by around 1% in performance. For Caltech101, OxfordPets, and Food101, our PLPP has significant improvements for all shots settings in two kind of prompts. The most obvious improvement is in Caltech101 dataset, 2 shot setting, PLPP aheads of 4.37% improvement for class-specific prompt.

### 4.3 BASE-TO-NEW GENERALIZATION

Given that CoOp exhibits a susceptibility to the issue of weak generalizability, leading to a substantial gap between accuracy on base classes and accuracy on unseen classes, CoCoOp introduces

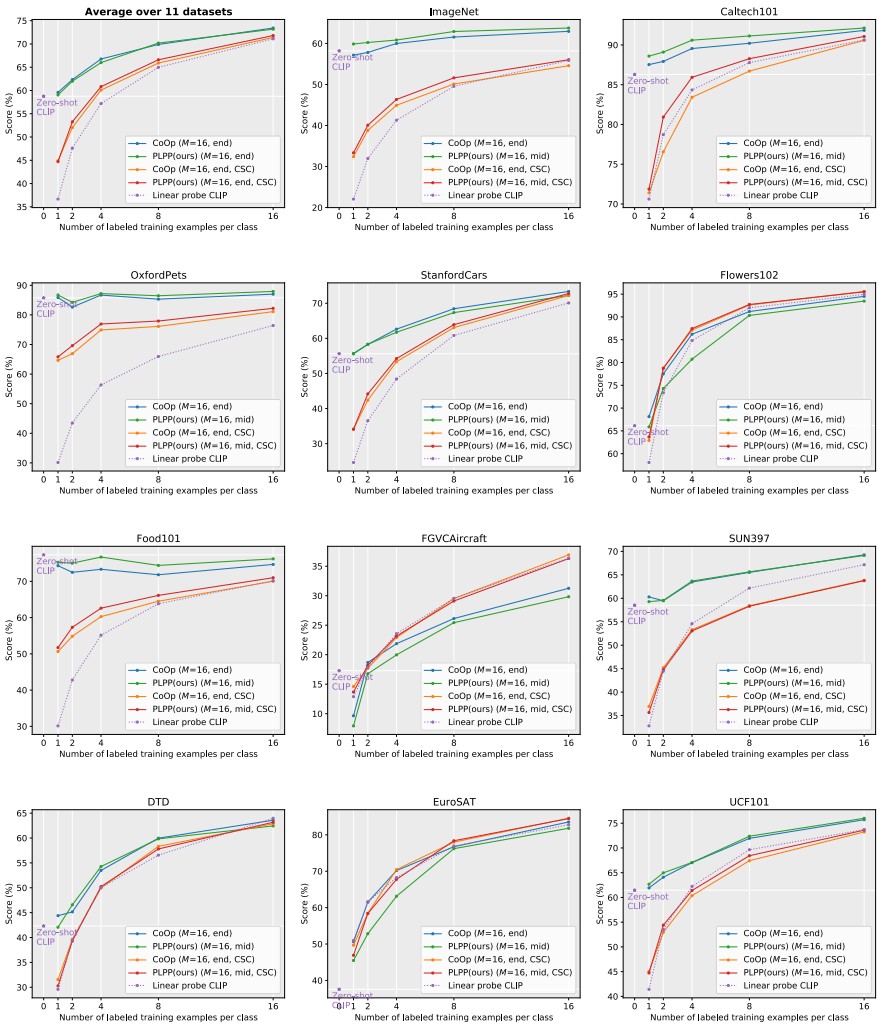

Figure 3: The few-shot learning results on 11 datasets. We comapre our PLPP with CoOp, and Zero-shot CLIP and observe the consistent and significant performance improvement on most datasets. (The average accuracy on all datasets is shown on the left top.)

image-conditioned mechanisms to address this limitation. In comparison to both CoOp and Co-CoOp, our PLPP adopts a calculation of cosine similarity to obtain the labels of vectors and an incremental LM head to integrate perplexity, and an identical network architecture, encompassing components such as prompts, text encoder, and image encoder. To assess the generalization performance from familiar classes to novel classes, we adopt a method of equal class division from CoCoOp, categorizing the classes into two distinct groups: base classes and new classes. All methods are exclusively trained on the base classes, with subsequent evaluation conducted on both base classes and new classes. Furthermore, we provide a composite measure by reporting the harmonic mean Xian et al. (2017) of accuracy scores for base classes and new classes, thereby facilitating a comprehensive assessment of the trade-off, where harmonic mean can be calculated as: $H = 2 * \frac{Base*New}{Base+New}$. As Table 1 shows, our PLPP and CoPLPP (CoCoOp + PLPP) achieve a certain lead on almost all data sets compared with CoOp Zhou et al. (2022b) and CoCoOp Zhou et al. (2022a).

Table 1: Accuracy (%) for the base-to-new generalization evaluation. The context length M is 16 for all methods and all methods are learned from the base classes with 16 shots. H: Harmonic mean. (CoPLPP is CoCoOp + PLPP)

(a) Average over 11 datasets.

|  | Base | New | H |
|---|---|---|---|
| CoOp | **83.19** | 62.04 | 71.07 |
| CoCoOp | 79.67 | 71.26 | 75.23 |
| PLPP | 82.49 | 63.00 | 71.44 |
| CoPLPP | 79.89 | **71.85** | **75.66** |

(b) Imagenet.

|  | Base | New | H |
|---|---|---|---|
| CoOp | **76.43** | 65.87 | 70.76 |
| CoCoOp | 76.24 | 70.87 | **73.46** |
| PLPP | 75.83 | 69.13 | 72.33 |
| CoPLPP | 75.73 | **70.97** | 73.27 |

(c) Caltech101.

|  | Base | New | H |
|---|---|---|---|
| CoOp | 97.83 | 90.40 | 93.97 |
| CoCoOp | 98.07 | **94.63** | 96.32 |
| PLPP | **98.17** | 91.40 | 94.66 |
| CoPLPP | 97.97 | **94.63** | 96.27 |

(d) OxfordPets.

|  | Base | New | H |
|---|---|---|---|
| CoOp | 93.87 | 92.53 | 93.20 |
| CoCoOp | **95.63** | 97.23 | 96.42 |
| PLPP | 94.6 | 92.93 | 93.76 |
| CoPLPP | 95.23 | **97.67** | **96.43** |

(e) StanfordCars.

|  | Base | New | H |
|---|---|---|---|
| CoOp | **80.17** | 56.53 | 66.31 |
| CoCoOp | 71.50 | 73.63 | 72.55 |
| PLPP | 76.70 | 58.50 | 66.38 |
| CoPLPP | 71.57 | **74.43** | **72.97** |

(f) Flowers102.

|  | Base | New | H |
|---|---|---|---|
| CoOp | **97.55** | 58.80 | 73.37 |
| CoCoOp | 92.67 | 71.43 | 80.68 |
| PLPP | 97.02 | 59.43 | 73.71 |
| CoPLPP | 92.43 | **72.13** | **81.03** |

(g) Food101.

|  | Base | New | H |
|---|---|---|---|
| CoOp | 87.93 | 83.77 | 85.80 |
| CoCoOp | 90.43 | 91.20 | 90.81 |
| PLPP | 88.87 | 85.77 | 87.29 |
| CoPLPP | **90.63** | **91.33** | **90.98** |

(h) FGVCAircraft.

|  | Base | New | H |
|---|---|---|---|
| CoOp | **43.13** | **22.63** | **29.68** |
| CoCoOp | 28.13 | 14.87 | 19.46 |
| PLPP | 40.63 | 15.73 | 22.68 |
| CoPLPP | 30.63 | 21.83 | 25.49 |

(i) SUN397.

|  | Base | New | H |
|---|---|---|---|
| CoOp | 80.26 | 61.60 | 69.70 |
| CoCoOp | 79.37 | 77.37 | 78.36 |
| PLPP | **80.53** | 64.33 | 71.52 |
| CoPLPP | 78.56 | **78.47** | **78.51** |

(j) DTD.

|  | Base | New | H |
|---|---|---|---|
| CoOp | **80.20** | 42.40 | 55.47 |
| CoCoOp | 75.97 | **57.03** | **65.15** |
| PLPP | 79.73 | 39.37 | 52.71 |
| CoPLPP | 76.37 | 55.67 | 64.40 |

(k) EuroSAT.

|  | Base | New | H |
|---|---|---|---|
| CoOp | **93.10** | 53.67 | 68.09 |
| CoCoOp | 86.53 | **61.00** | **71.56** |
| PLPP | 91.73 | 48.47 | 63.43 |
| CoPLPP | 87.37 | 58.40 | 70.01 |

(l) UCF101.

|  | Base | New | H |
|---|---|---|---|
| CoOp | **84.63** | 54.20 | 66.08 |
| CoCoOp | 81.83 | 74.60 | 78.05 |
| PLPP | 83.63 | 67.93 | 74.97 |
| CoPLPP | 82.30 | **74.80** | **78.37** |

## 4.4 DOMAIN GENERALIZATION

The domain generalization paradigm assesses the models' capacity for generalization in a target domain distinct from the source domain. Traditional fine-tuning with a limited dataset from a specific domain can potentially mislead the model into acquiring spurious correlations or patterns confined to that domain, thus yielding a biased model that exhibits subpar performance in unfamiliar domains. Given that our PLPP leverages perplexity to supervise the training process of prompts, our PLPP demonstrates resilience in the face of distributional shifts. As exemplified in Table 2, our experimetal results demonstrate that PLPP consistently outperforms CoOp across all target datasets. Furthermore, CoPLPP, even achieves superior performance compared to CoCoOp in all target datasets.

## 4.5 VISUALIZING LEARNED PROMPTS

Since PLPP is a tentative work to make learned prompt more comprehensible, we visualiaze the prompt through calculating the Euclidean distance between learned prompt and $embedding.weight$, return the index of minimum distance. Then we use tokenizer from CLIP to convert the indexs into corresponding words. Finally, we show the converted words in Table 3. To demonstrate superiority of our method, we also show the converted words trained by CoOp to make a evident comparison. From Table 3 we can know that compared with CoOp, our method can make the learned prompt more comprehensible, i.e., actual existing words have increased. To minimize chance occurrence, in few-shot classification of 11 datasets, we count the number of actual existing words in all learned

Table 2: Comparison of PLPP and CoPLPP with existing approaches in domain generalization setting. The context length M is 16 for all methods and the prompts are trained with 2 shots. PLPP and CoPLPP show certain improvements on the target datasets.

| | Source | Target | | | |
|---|---|---|---|---|---|
| | ImageNet | ImageNetV2 | ImageNet-Sketch | ImageNet-A | ImageNet-R |
| CoOp | 67.60 | 60.4 | 44.9 | 47.57 | 72.23 |
| CoCoOp | 70.20 | 63.57 | 48.60 | 50.93 | 76.03 |
| PLPP | 68.60 | 61.83 | 47.07 | 49.50 | 75.43 |
| CoPLPP | **70.23** | **63.7** | **48.73** | **51.07** | **76.47** |

class specific prompts, by leaveraging third-party Python libraries. Statistics indicate that prompts (M=16) trained by our PLPP containing actual words is about 2.3 more than CoOp on average. However, although there has been a certain improvement in comprehensibility. Overall, the learned prompts still have a long way to go from hand-crafted prompts.

Table 3: Visualizing learned prompts trained by CoOp and our PLPP, the left prompts are trained by CoOp and the right are trained by PLPP

| # | ImageNet | Food101 | OxfordPets |
|---|---|---|---|
| 1 | potd | lc | tosc |
| 2 | that | enjoyed | judge |
| 3 | filmed | beh | fluffy |
| 4 | fruit | matches | cart |
| 5 | . | nytimes | harlan |
| 6 | ° | prou | paw |
| 7 | excluded | lower | incase |
| 8 | cold | N/A | bie |
| 9 | stery | minute | snuggle |
| 10 | warri | ~ | along |
| 11 | marvelcomics | well | enjoyment |
| 12 | .: | ends | jt |
| 13 | N/A | mis | improving |
| 14 | lation | somethin | srsly |
| 15 | muh | seminar | asteroid |
| 16 | .# | N/A | N/A |

| # | ImageNet | Food101 | OxfordPets |
|---|---|---|---|
| 1 | priority | civilian | delavin |
| 2 | verified | muddy | drunk |
| 3 | cruelty | oy | discipl |
| 4 | encounters | roasted | tortured |
| 5 | ians | me | sentences |
| 6 | lips | scup | crayon |
| 7 | targets | lower | – |
| 8 | waitrose | bel | reception |
| 9 | ounded | cocac | stick |
| 10 | conveni | ..# | onents |
| 11 | cole | helle | metrical |
| 12 | ga | ounding | misunderstood |
| 13 | answer | chance | executive |
| 14 | whi | bebe | finalized |
| 15 | build | seminar | agricultural |
| 16 | ) | km | smallest |

# 5 CONCLUSION

In this paper, we elucidate the predicaments prevailing CoOp-based methods that learned prompts contain many non-existent words, resulting incomprehensible. These methods have exclusively concentrated on improving performance across various downstream tasks, while overlook the incomprehensible resulting prompts. Thus, we introduce a novel prompt-tuning technique, PLPP, which can be integrated into any CoOp-based methods. PLPP imposes the evaluation metric perplexity in the training process of prompts to make resulting prompts more comprehensible. In order to integrate perplexity in the training process, we utilize cosine similarity to obtain the labels of learnable vectors and add a LM head to output word probability distribution. Empirical assessments are conducted across diverse domains, including few-shot classification, base-to-new generalization, and domain generalization encompassing 11 distinct datasets, have unequivocally underscored the slight improvement in the performance of our PLPP method. We also visualize the prompt trained by PLPP and CoOp, which indicates that the former prompts are more comprehensible. Looking forward, since there are still some differences between our prompt and hand-crafted prompt, we envision further exploration into more potent training paradigm and ultimately realize our original intention to facilitate applying VL model to safety critical scenarios.

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
