# A APPENDIX

---

**Algorithm 1** PLPP
---
1: **function** BATCH-UPDATING($I, Prompts$)
2:     # Get The Features of the current batch, labels and output distribution of prompts.
3:     $I_f \leftarrow image\_encoder(I)$
4:     $T_f \leftarrow text\_encoder(Prompts)$
5:     $\tilde{T}_f \leftarrow text\_encoder\_before\_proj(Prompts)$
6:     $prompts\_label \leftarrow generate\_label(prompts)$
7:     $output \leftarrow lm\_head(\tilde{T}_f)$
8:
9:     # Calculate the losses.
10:     $L_{CE} \leftarrow \text{CE}(I_f @ T_f.transpose(), label)$
11:     $L_{PPL} \leftarrow$ TEXT-PPL-LOSS($output, prompts\_label$)
12:     $L_{PLPP} \leftarrow L_{CE} + \lambda L_{PPL}$
13:
14:     # Update the prompts.
15:     $Prompts \leftarrow$ BACKWARD-UPDATE($output, prompts\_label$)
16: **end function**
17:
18: **function** TEXT-PPL-LOSS($output, prompts\_label$)
19:     $loss \leftarrow CE(output, prompts\_label)$
20:     $ppl \leftarrow e^{loss}$
21:     **return** $ppl$
22: **end function**

---

Table 4: Visualizing prompts trained by our PLPP

| #  | UCF101      | Flowers102 | Caltehc101 | StanfordCars | FGVCAircraft |
|----|-------------|------------|------------|--------------|--------------|
| 1  | verify      | hooper     | learning   | headquarters | humor        |
| 2  | exhilarating| hike       | 6          | minis        | discuss      |
| 3  | igne        | beh        | shoot      | rebound      | existed      |
| 4  | discuss     | accurate   | reduces    | ator         | city         |
| 5  | gi          | core       | uta        | sport        | pakistan     |
| 6  | fx          | signing    | 2          | ville        | direct       |
| 7  | visited     | vs         | inspired   | stats        | imma         |
| 8  | night       | iq         | fields     | machine      | ays          |
| 9  | real        | manager    | cartoon    | back         | injury       |
| 10 | drop        | per        | mark       | amount       | womensday    |
| 11 | controls    | hike       | bubble     | drew         | ho           |
| 12 | suggestions | website    | hike       | sa           | hel          |
| 13 | bec         | year       | babe       | 2            | lemans       |
| 14 | smile       | und        | threatened | herself      | its          |
| 15 | edge        | 2          | shoutout   | version      | kick         |
| 16 | jersey      | seventeen  | cs         | colored      | papa         |