# OpenReview forum: "PLPP: PROMPT LEARNING WITH PERPLEXITY FOR VISION-LANGUAGE MODELS"
_ICLR.cc/2024/Conference — ICLR 2024 Conference Withdrawn Submission_

### Official Review · Reviewer_NG2C · 2023-10-21

**Soundness:** 2 fair
**Presentation:** 3 good
**Contribution:** 2 fair
**Rating:** 3
**Confidence:** 5

**Summary:**

This paper introduces Perplexity as a supervisory signal for guiding the prompt learning process within the CoOp framework, aiming at the generation of comprehensible prompts. The proposed methodology involves a two-step operation to assess perplexity for prompts. In the initial step, cosine similarity is computed to derive vector labels, followed by a subsequent step utilizing a training-free language model head to generate the probability distribution for words.

**Strengths:**

${\bf Strengths:}$

[$\textbf{Simple Plug-in}$] The proposed PLPP is a simple plug-in that can be integrated into any CoOp-based method.

[$\textbf{Interesting Problem}$] The work finds and focuses on an interesting problem.

**Weaknesses:**

${\bf Weaknesses:}$

[$\textbf{Limited Novelty}$] The only technical contribution of this work is to introduce an existing metric for perplexity to the CoOp framework.

[$\textbf{Limited Applied Scenarios}$] It seems the work is specialized to solve the problem of CoOp based methods, which limits its applicability.

[$\textbf{Some Confusions}$] Some expressions used in writing may cause confusion. For example, “CLIP’s training regimen”, “in Figure 2 whether only learnable vectors are used for obtaining the label?”. Also, it is not clear why use the “embedding.weight” to label the prompts.

[$\textbf{Missing Related Works}$] Some adapter-style tuning works are not discussed, e.g., “Prompt, Generate, then Cache: Cascade of Foundation Models makes Strong Few-shot Learners, CVPR2023” and “Task Residual for Tuning Vision-Language Models, CVPR2023”.

[$\textbf{Missing Comparisons with SOTA Methods}$] “MaPLe: Multi-modal Prompt Learning, CVPR2023” and “Self-regulating Prompts: Foundational Model Adaptation without Forgetting, ICCV2023” are SOTA methods that adopt prompt learning. It would be nice to compare the generated prompts with these methods and the performance of these methods by adding the proposed loss in this method.

[$\textbf{Non-convincing Prompts}$] The visualization in Table 3 does not show the advantage of the proposed method. We can see this in three perspectives: (i) the generated prompts by PLPP are not always related to the dataset, e.g., “verified” for ImageNet, “chance” for Food101 and “crayon” for OxfordPets; (ii) CoOp sometimes generates more accurate prompts, e.g., “fruit” for ImageNet, “enjoyed” for Food101 and “paw” for OxfordPets; (iii) it seems the method does not work sometimes as prompts are same for # 15 on  Food101.

[$\textbf{Some Typos}$] There are some typos to be fixed, e.g., “A OVERVIEW”, “used to aseess” and “leanable vectors”. Please do careful proofreading before submission.

[$\textbf{Disordered Reference Formats}$] The format of reference should be organized. For example, the authors can choose one of “European Conference on Computer Vision” and “ECCV” to make it neat.

**Questions:**

Please refer to the weaknesses.

---

> ### Author Response · Authors · 2023-11-17
>
> Thanks for your constructive comments. We provide clarification to your questions and concerns as below. If you have any further questions or comments, please post them and we will be happy to have further discussions. We will keep to explore our method to achieve satisfactory (completely comprehensible) experimental results.
>
> **Q1**: Limited Novelty.
>
> **A1**: Novelty should not be limited by technical contribution, but motivation, technical contribution, and experimental results. For example, CoOp, which (only) uses learnable prompt instead of hand-crafted prompt. We will keep to explore our method to achieve satisfactory (completely comprehensible) experimental results.
>
> **Q2**: Limited Applied Scenarios.
>
> **A2**: The key of CoOp based methods is introducing prompt learning from NLP, and in Multimodal Large Language Model (MLLM), prompts are called learned queries as part of input of Large Language Model. Our method can also used in the training the learned queries of MLLM.
>
> **Q3**: Some Confusions.
>
> **A3**: We will use more clear expression to reduce confusion. Learnable vectors have no labels, so We need to get the labels to facilitate subsequent calculation of PPL loss, and we can directly get the index (label) of {classname} and ‘.’according to vocab. Normally, when a sentence is sent to the language model, each word in the sentence is first mapped into an index according to the vocab, and then use the word embedding layer (nn.Embedding()) to convert index to vector. The“embedding.weight”is to convert index (label) into the corresponding vector, so we use the“embedding.weight”to label the prompts.
>
> **Q4**: Missing Related works.
>
> **A4**: Thank you for listing the related adapter-style tuning works, we will add them in the related works of my paper.
>
> **Q5**: Missing comparisons with SOTA methods.
>
> **A5**: We will conduct comparative experiments with the two methods. Due to time constraints, we are unable to complete the experiment within the deadline.
>
> **Q6**: Non-convincing prompts.
>
> **A6**: We will keep to explore our method to achieve satisfactory (completely comprehensible) experimental results.
>
> **Q7**: Typos and disordered reference format.
>
> **A7**: Thank you, we will fix the mentioned writing error and unify the reference format.

---

### Official Review · Reviewer_42yP · 2023-10-22

**Soundness:** 2 fair
**Presentation:** 2 fair
**Contribution:** 1 poor
**Rating:** 3
**Confidence:** 5

**Summary:**

This paper aims to improve prompting learning for VLMs from the perspective of learning comprehensible prompts. Based on CoOp/CoCoOp, it uses an extra perplexity metric to optimize prompt learning. The experiments are conducted on common task settings, including few-shot classification, base-to-new generalization, and domain generalization.

**Strengths:**

- The idea of learning comprehensible prompts for VLMs is new, and this paper serves as the first attempt to explore that.
- The experiments are conducted in multiple task settings.

**Weaknesses:**

- The prompts learned by the proposed method are not very comprehensible. As shown in Table 3, even though the number of actual existing words has increased compared to CoOp, the prompts themselves lack comprehensibility. However, the primary motivation of this paper is to learn comprehensible prompts, and it appears that this goal has not been achieved by the method.
- On the other hand, the performance of the proposed method is only comparable with CoOp and notably underperforms the state-of-the-art (SOTAs). On some datasets, it even performs worse than CoOp, such as Flowers102, FGVCAircraft, EuroSAT. What could be the reasons for this? Although the focus of this paper is not on performance, both its performance and the focus on comprehensibility have not shown significant improvements. The experimental evaluations do not seem to validate the claimed contributions.
- The perplexity metric is used to improve the comprehensibility. However, the perplexity metric cannot explicitly guide the prompts to be comprehensible.
- This paper claims that “readable prompts help in understanding the prediction results and applying the VL models in safety-critical scenarios.”. However, the readability does not necessarily imply safety. The human-comprehensible prompt may not be the best choice for VLMs.
- As “PLPP can be integrated in any CoOp-based method”, does it also validate useful for SOTA, rather than for the baselines CoOp and CoCoOp?

**Questions:**

See the weaknesses for details.

---

> ### Author Response · Authors · 2023-11-17
>
> Thanks for your constructive comments. We provide clarification to your questions and concerns as below. If you have any further questions or comments, please post them and we will be happy to have further discussions.
>
> **Q1**: The prompts learned by the proposed method are not very comprehensible.
>
> **A1**: Yes, even if actual existing words increase, the prompts learned by our method is still incomprehensible not like human-writing sentences. We will continue to work towards making the prompt fully comprehensible.
>
> **Q2**: On the other hand, the performance of the proposed method is only comparable with CoOp and notably underperforms the state-of-the-art (SOTAs). On some datasets, it even performs worse than CoOp, such as Flowers102, FGVCAircraft, EuroSAT. What could be the reasons for this?
>
> **A2**: The performance of our method on Flowers102 only worse in Base class. As for the other two datasets FGVCAircraft and EuroSAT, I think the main reason is that the two dataset are very sensitive to the prompts, and compared to the loss related to accuracy, the weight of PPL loss in final loss is too large resulting this circumstance.
>
> **Q3**: The perplexity metric is used to improve the comprehensibility. However, the perplexity metric cannot explicitly guide the prompts to be comprehensible.
>
> **A3**: According to the definition of perplexity, I think perplexity metric has the hope of guiding the prompts to be comprehensible, but some other help is needed, and we will keep explore to achieve this goal.
>
> **Q4**: This paper claims that “readable prompts help in understanding the prediction results and applying the VL models in safety-critical scenarios.”. However, the readability does not necessarily imply safety. The human-comprehensible prompt may not be the best choice for VLMs.
>
> **A4**: Readability does not mean absolute safety, but it can to some extent convince us of the model's prediction results. As for incomprehensible prompts, how can it make people trust the model's predictions in safety-critical areas. Just like using some explainable methods such as GradCAM, GradCAM++ to show the reasons for model predictions. An image about a dog sit on the grass, the model predict the image as dog, however, the reasons of the predicted result are all on grass, people will believe the reasons on dog rather than on grass.
>
> **Q5**: As “PLPP can be integrated in any CoOp-based method”, does it also validate useful for SOTA, rather than for the baselines CoOp and CoCoOp?
>
> **A5**: Yes, PLPP can be integrated in any CoOp-based method. We have been thought about using PLPP on MaPLE, but MaPLE not only has prompts in text input, but also in vision encoder and in intermediate layer, which conflicts with our motivation. We will conduct the comparative experiments only on performance.

---

### Official Review · Reviewer_6oUS · 2023-10-31

**Soundness:** 1 poor
**Presentation:** 2 fair
**Contribution:** 1 poor
**Rating:** 1
**Confidence:** 4

**Summary:**

This paper presents a method to improve prompt learning of VLMs by adding a perplexity loss to the learnable prompts. The PPL loss is computed using the word distributions obtained by feeding the outputs of the CLIP text encoder to the transpose of the word embedding layer. Experiments show the method can improve both CoOp and CoCoOp baselines.

**Strengths:**

- The method is easy to understand.

**Weaknesses:**

- The implementation of using PPL to supervise prompt learning is not convincing and reasonable. PPL is used to evaluate LMs as the outputs of the LMs model the probability distribution over a predefined vocabulary of words. According to the implementation details provided in Section 3.3, the authors obtain the word probability distribution of the input prompts by feeding the outputs of the CLIP text encoder to the transpose of the word embedding layer. The output of the CLIP text encoder is designed to compute the similarity between texts and images, instead of modeling the word distribution as LMs like BERT and GPT.  I think there is no guarantee that the output models the word probability distribution. Therefore, it is not meaningful to optimize the PPL of the distribution, which means the core design of the proposed method is problematic. How about using a pre-trained language model to calculate the PPL loss?

- The experimental results are weak. The improvements reported in Table 1 are quite marginal and are likely within the random errors. According to results reported in MaPLe, the average performance of CoOp and CoCoOp are 71.66 and 75.83 respectively, which is higher than both the baseline results reported in this paper (71.07 and 75.23) and the proposed method (71.44 and 75.66). It is possible that the proposed method is not effective at all.

- The motivation of the method is not well-supported by the results. The paper emphasizes that it is "the first work aiming at learning comprehensible prompts". However, according to Table 3, the learned prompts are also incomprehensible.

**Questions:**

Considering the questionable designs of the proposed method, the weak experimental results, and the unclear motivation, I cannot recommend acceptance for the paper. I think most of the critical issues cannot be easily solved through rebuttal, I would like to rate "strong reject" for this paper.

----- post rebuttal ----
After reading the authors' feedback as well as other reviews, I would like to keep my initial rating. My main concerns about the questionable design of the proposed method and experimental results have not been addressed.

---

> ### Author Response · Authors · 2023-11-17
>
> Thanks for your constructive comments. We provide clarification to your questions and concerns as below. If you have any further questions or comments, please post them and we will be happy to have further discussions.
>
> **Q1**: The proposed method is problematic. How about using a pre-trained language model to calculate the PPL loss?
>
> **A1**: First of all, the CLIP text encoder is to output the feature of input text. During pretraining, image-captions pair data are fed into the image encoder and text encoder, aligning the feature space of outputs. The pretraining stage will shift the feature space of the output from the text encoder, while simultaneously optimizing the parameters of the embedding layer (nn.Embedding()), so we can still model the word probability distribution by the output of text encoder and the word embedding layer. Secondly, I have thought about using a pre-trained language model to calculate the PPL loss, however, different language model have different vocab, which means that even if the prompt has been optimized by PPL loss and Bert to be very comprehensible, in the view of CLIP text encoder, the prompt is still incomprehensible. Furthermore, I have investigated all the language models, and none of them are the same as the vocab CLIP text encoder uses. Finally, I will conduct some experiments by using a pre-trained language model to calculate the PPL loss, and observe the difference of result.
>
> **Q2**: The experimental results are weak. Different baseline results compared with MaPLE.
>
> **A2**: According to Table 1 in our paper, PLPP has a significant improvement compared to CoOp on almost all datasets except FGVCAircraft and EuroSAT, causing a less significant increase in average. As for the reasons for the difference in average performance about CoOp and CoCoOp, I think different setting on context length causing the results. In our method, the context length is set to 16 and I do not see the information about the context length of CoOp and CoCoOp in MaPLE. Since our method can be integrated into any CoOp-based method, we can also integrate into MaPLE, and we will conduct the comparative experiments.
>
> **Q3**: The motivation of the method is not well-supported by the results.
>
> **A3**: Although actual existing words increase in learned prompts, learn prompts are indeed incomprehensible not like human-writing sentences. I will continue to work towards making the prompt comprehensible like human-writing sentences.

---

### Official Review · Reviewer_hPSs · 2023-11-10

**Soundness:** 2 fair
**Presentation:** 2 fair
**Contribution:** 2 fair
**Rating:** 3
**Confidence:** 5

**Summary:**

This paper studies the incomprehensible prompt issue in current text-image alignment models. It designs a cross-entropy term to detect the perplexity of prompts. Based on that, they use this metric to optimize plausible prompts instead of tuning the model. Their method demonstrates power in fixing the perplexity issue. Many cases show that their method can find the prompt with appropriate meanings. The authors also demonstrate this learning paradigm will not hurt the ability of the model, and witness improvement of performance in some cases.

**Strengths:**

1. The idea of this paper is novel, they observe the perplexity issue it current language models and try to fix it from the text prompts level instead of tuning the model directly. This means the model will not lose too much capacity due to tuning but can also convey readable information to users.

2. The visualization method is very interesting and shows the great potential of this method. However if they authors can do a large-scale user study of this phenomenon, and produce numerical metrics for readers, this paper will be much more convincing. Currently, I just do not know to what extent this method can fix the perplexity issue.

3. Very extensive study on classification scenarios.

**Weaknesses:**

Major concerns:
1. The experiment results are somewhat marginal:
   in the domain generalization case, PLPP outperforms CoOp significantly, but when extended to CoPLPP and CoCoOp, the difference is marginal.
   in the base2new setting, the results are barely compatible with CoOp and CoPLPP, in StanfordCars, the CoOp outperforms your method by 9%, such cases are frequently seen among the base2new results in both base, new, and H categories, which is way too impressive.

2. I think a weak score is not the main reason to reject a paper, provided it provides new insight for readers. While the inducing of the perplexity score is interesting. The authors did not deeply study this metric. First of all, you can give an analysis of how many handwriting prompts we need in order to train a perplexity model. You can also plot or simply list the change of prompts as you optimize the perplexity score, to let us see how your method drives the prompts to be more and more feasible to read. Besides, no ablation about lambda is introduced in your experiment setting, we do not even know the role of CE and PPL in your learning, which will be dominating in improving the perplexity? How is the influence? Is larger lambda better? What is the trade-off when we improve the perplexity? Why did you cross Entropy, why not L2 loss? I believe there is much space for you to keep explore to make this paper with more depth.

3. How will this method influence the retrieving ability and other abilities of the CLIP model?

4. No experiments in other CLIP tasks, making this work a bit narrow.




some minor issues:
1. in summary, the third point, an extra dot before PLPP
2. Be careful to distinguish probability and probability density.
3. What are t_i and t_j in eq 1? Are they p_i and p_j?
4/ . It’s worth noting that in the realm of Natural Language Processing (NLP), perplexity is
primarily employed to evaluate the efficacy of a language model when exposed to human-authored
sentences. Could you kindly provide me with some references here?
5. In the implementation part, could you tell me which language model you use?
6. Now that you report the mean of three runs, there should be std in your figures.
7. no , after each equation. You need to add punctuation after equations, they are parts of sentences.
8. H = 2 * Base *New / Base + Mean, we usually use \cdot or \times in this case, * usually denote the convolution operator.

**Questions:**

1. Do you heavily use ChatGPT to polish this paper? I think some of its writing styles are not good for reading fluency, I prefer those in the book 'Elements of Style'. I prefer simple sentences as clear subject + verb + object, not short sentences followed by long causes and long causes. It's really difficult to read if full of such sentences. I am not a litterateur I cannot judge its elegance. But as a researcher, I am not a fan of this style.  This is just for discussion, not related to the rating. Be relax. I hope you find the book 'Elements of Style' useful.

2. I recommend you follow the ICLR official guidelines for notations. In the guideline pdf and corresponding tex file, at the end page, there is a list of how to use math notations. Simply say, $\bm{v}$ for vectors, $v$ for scalars, $\bm{V}$ for matrices, $\mathrm{PPL}$ (\mathrm for text in a notation, you may find that document to learn the remains. It is really helpful to write equitation and notations like that, much more helpful for readers to get what you say.

3. Note your citation style, use \cite, \citep, \citeauthor wisely based on what you want to cite, a method, a people or a supplementary to something.

4. Note your reference style, if you decided to use abbreviation， use it consistently in all references. Do not use The IEEE/CVF Conference on Computer Vision and Pattern Recognition 2023 together with CVPR in your references.

---

> ### Author Response · Authors · 2023-11-17
>
> Thanks for your constructive comments. We provide clarification to your questions and concerns as below.
>
> **Q1**: There is much space for the author to keep explore to make this paper with more depth.
>
> **A1**: Thank you for pointing out the weaknesses of this paper and your suggestion about the experiments, and your experimental suggestions are very enlightening. We will keep to explore.
>
> **Q2**: Do you heavily use ChatGPT to polish this paper?
>
> **A2**: Yes, I heavily used ChatGPT to polish this paper, and I thought high-level sentences would be more attractive. However, Writing a paper should be aimed at letting readers know what I did and how I did it, I made a mistake. Thank you for recommending the book ‘Elements of Style’.
>
> **Q3**: Notations, citation style and reference style.
>
> **A3**: Thank you! I will follow ICLR official guidelines to correct my notations, correct my citation according to what I want to cite and unify my reference style.
>
> **Q4**: Some minor issues.
>
> **A4**: Thank you! We will fix all the issues.